# Model Adaptation: Historical Contrastive Learning for Unsupervised Domain Adaptation without Source Data

**Jiaxing Huang, Dayan Guan, Aoran Xiao, Shijian Lu**[*]
School of Computer Science Engineering, Nanyang Technological University
{Jiaxing.Huang, Dayan.Guan, Aoran.Xiao, Shijian.Lu}@ntu.edu.sg

## Abstract

Unsupervised domain adaptation aims to align a labeled source domain and an unlabeled target domain, but it requires to access the source data which often raises concerns in data privacy, data portability and data transmission efficiency. We study unsupervised model adaptation (UMA), or called Unsupervised Domain Adaptation without Source Data, an alternative setting that aims to adapt source-trained models towards target distributions without accessing source data. To this end, we design an innovative historical contrastive learning (HCL) technique that exploits historical source hypothesis to make up for the absence of source data in UMA. HCL addresses the UMA challenge from two perspectives. First, it introduces historical contrastive instance discrimination (HCID) that learns from target samples by contrasting their embeddings which are generated by the currently adapted model and the historical models. With the historical models, HCID encourages UMA to learn instance-discriminative target representations while preserving the source hypothesis. Second, it introduces historical contrastive category discrimination (HCCD) that pseudo-labels target samples to learn category-discriminative target representations. Specifically, HCCD re-weights pseudo labels according to their prediction consistency across the current and historical models. Extensive experiments show that HCL outperforms and state-of-the-art methods consistently across a variety of visual tasks and setups.

## 1 Introduction

Deep neural networks (DNNs) [28, 73, 23] have achieved great success in various computer vision tasks [8, 55, 60, 59, 28, 73, 23] but often generalize poorly to new domains due to the inter-domain discrepancy [1]. Unsupervised domain adaptation (UDA) [78, 51, 76, 64, 66, 79, 77, 103, 102, 25, 71, 44, 26, 86] addresses the inter-domain discrepancy by aligning the source and target data distributions, but it requires to access the source-domain data which often raises concerns in data privacy, data portability, and data transmission efficiency.

In this work, we study unsupervised model adaptation (UMA), an alternative setting that aims to adapt source-trained models to fit target data distribution without accessing the source-domain data. Under the UMA setting, the only information carried forward is a portable source-trained model which is usually much smaller than the source-domain data and can be transmitted more efficiently [45, 42, 43, 72, 48] as illustrated in Table 1. Beyond that, the UMA setting also alleviates the concern of data privacy and intellectual property effectively. On the other hand, the absence of the labeled source-domain data makes domain adaptation much more challenging and susceptible to collapse.

---

[*]Corresponding author.

35th Conference on Neural Information Processing Systems (NeurIPS 2021).

Table 1: Source data have much larger sizes than source-trained models.

| Storage size (MB) | Semantic segmentation | | Object detection | Image classification |
|---|---|---|---|---|
| | GTA5 | SYNTHIA | Cityscapes | VisDA17 |
| Source dataset | $62,873.6$ | $22,323.2$ | $12,697.6$ | $7,884.8$ |
| Source-trained model | $179.1$ | $179.1$ | $553.4$ | $172.6$ |

To this end, we develop historical contrastive learning (HCL) that aims to make up for the absence of source data by adapting the source-trained model to fit target data distribution without forgetting source hypothesis, as illustrated in Fig. 1. HCL addresses the UMA challenge from two perspectives. First, it introduces historical contrastive instance discrimination (HCID) that learns target samples by comparing their embeddings generated by the current model (as queries) and those generated by historical models (as keys): a query is pulled close to its positive keys while pushed apart from its negative keys. HCID can thus be viewed as a new type of instance contrastive learning for the task of UMA with historical models, which learns instance-discriminative target representations without forgetting source-domain hypothesis. Second, it introduces historical contrastive category discrimination (HCCD) that pseudo-labels target samples for learning category-discriminative target representations. Specifically, HCCD re-weights the pseudo labels according to their consistency across the current and historical models.

The proposed HCL tackles UMA with three desirable features: 1) It introduces historical contrast and achieves UMA without forgetting source hypothesis; 2) The HCID works at instance level, which encourages to learn instance-discriminative target representations that generalize well to unseen data [98]; 3) The HCCD works at category level (*i.e.*, output space) which encourages to learn category-discriminative target representation that is well aligned with the objective of down-stream tasks.

The contributions of this work can be summarized in three aspects. *First*, we investigate memory-based learning for unsupervised model adaptation that learns discriminative representations for unlabeled target data without forgetting source hypothesis. To the best of our knowledge, this is the first work that explores memory-based learning for the task of UMA. *Second*, we design historical contrastive learning which introduces historical contrastive instance discrimination and category discrimination, the latter is naturally aligned with the objective of UMA. *Third*, extensive experiments show that the proposed historical contrastive learning outperforms state-of-the-art methods consistently across a variety of visual tasks and setups.

## 2 Related Works

Our work is closely related to several branches of research in unsupervised model adaptation, domain adaptation, memory-based learning and contrastive learning.

**Unsupervised model adaptation** aims to adapt a source-trained model to fit target data distributions without accessing source-domain data. This problem has attracted increasing attention recently with a few pioneer studies each of which focuses on a specific visual task. For example, [45, 46] freezes the classifier of source-trained model and performs information maximization on target data for classification model adaptation. [42] tackles classification model adaptation with a conditional GANs that generates training images with target-alike styles and source-alike semantics. [43] presents a self-entropy descent algorithm to improve model adaptation for object detection. [72] reduces the uncertainty of target predictions (by source-trained model) for segmentation model adaptation. [48] introduces data-free knowledge distillation to transfer source-domain knowledge for segmentation model adaptation. Despite the different designs for different tasks, the common motivation of these studies is to make up for the absence of source data in domain adaptation. [40] and [88] tackle source-free domain adaptation from a generative manner by generating samples from the source classes and generating reference distributions, respectively.

We tackle the absence of source data by a memory mechanism that encourages to memorize source hypothesis during model adaptation. Specifically, we design historical contrastive learning that learns target representations by contrasting historical and currently evolved models. To the best of our knowledge, this is the first work that explores memory mechanism for UMA.

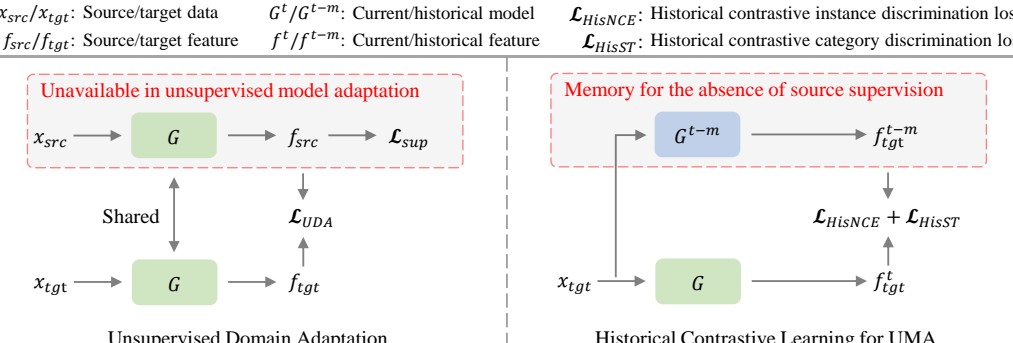

Figure 1: Illustration of unsupervised domain adaptation, unsupervised model adaptation and the proposed historical contrastive learning which exploits historical source hypothesis (or memorized knowledge) to make up for the absence of source supervision in the process of UMA. Here the historical source hypothesis could be the original source hypothesis $G^0$ (*i.e.* t=m, trained using the labeled source data only), the adapted source hypothesis $G^{t-m}$ (*i.e.* $m < t$, trained in the last $m$ epoch), or other types of previous models.

**Domain adaptation** is related to UMA but it requires to access labeled source data in training. Most existing work handles UDA from three typical approaches. The first exploits *adversarial training* to align source and target distributions in the feature, output or latent space [78, 51, 76, 14, 96, 64, 66, 79, 34, 77, 35, 21, 94, 29, 10, 9, 80, 20]. The second employs *self-training* to generate pseudo labels to learn from unlabeled target data iteratively [103, 69, 100, 102, 31, 19, 90, 91]. The third leverages *image translation* to modify image styles to reduce domain gaps [25, 71, 44, 95, 26, 86, 32, 33, 93, 92]. In addition, [30] proposes a categorical contrastive learning for domain adaptation.

**Memory-based learning** has been studied extensively. Memory networks [81] as one of early efforts explores to use external modules to store memory for supervised learning. Temporal ensemble [41], as well as a few following works [74, 12] extend the memory mechanism to semi-supervised learning. It employs historical hypothesis/models to regularize the current model and produces stable and competitive predictions. Mean Teacher [74] leverages moving-average models as the memory model to regularize the training, and similar idea was extended for UDA [16, 99, 4, 52]. Mutual learning [97] has also been proposed for learning among multiple peer student models.

Most aforementioned methods require labeled data in training. They do not work very well for UMA due to the absence of supervision from the labeled source data, by either collapsing in training or helping little in model adaptation performance. We design innovative historical contrastive learning to make up for the absence of the labeled source data, more details to be presented in the ensuing subsections.

**Contrastive learning** [82, 87, 22, 54, 101, 24, 56, 75, 36, 11] learns discriminative representations from multiple views of the same instance. It works with certain dictionary look-up mechanism [22], where a given image $x$ is augmented into two views, query and key, and the query token $q$ should match its designated key $k_+$ over a set of negative keys $k_-$ from other images. Existing work can be broadly classified into three categories based on dictionary creation strategies. The first creates a *memory bank* [82] to store all the keys in the previous epoch. The second builds an *end-to-end* dictionary [87, 75, 36, 11] that generates keys using samples from the current mini-batch. The third employs a *momentum encoder* [22] that generates keys on-the-fly by a momentum-updated encoder.

**Other related source-free adaptation works.** [7] considers supervised continual learning from previously learned tasks to a new task, which learns representations using the contrastive learning objective and preserves learned representations using a self-supervised distillation step, where the contrastively learned representations are more robust against the catastrophic forgetting for supervised continual learning. [38] addresses a source-free universal domain adaptation problem that does not guarantee that the classes in the target domain are the same as in the source domain. [39] propose a simple yet effective solution to realize inheritable models suitable for open-set source-free DA problem.

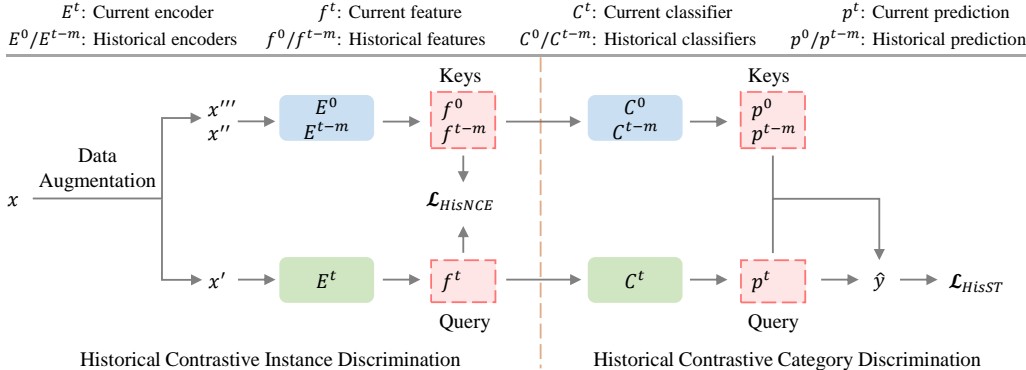

Figure 2: The proposed historical contrastive learning consists of two key designs including historical contrastive instance discrimination (HCID) and historical contrastive category discrimination (HCCD). HCID learns from target samples by contrasting their embeddings generated by the current model (as queries) and historical models (as keys), which learns instance-discriminative target representations. HCCD pseudo-labels target samples to learn category-discriminative target representations, where the pseudo labels are re-weighted adaptively according to the prediction consistency across the current and historical models.

## 3 Historical Contrastive Learning

This section presents the proposed historical contrastive learning that memorizes source hypothesis to make up for the absence of source data as illustrated in Fig. 2. The proposed HCL consists of two key designs. The first is historical contrastive instance discrimination which encourages to learn instance-discriminative target representations that generalize well to unseen data [98]. The second is historical contrastive category discrimination that encourages to learn category-discriminative target representations which is well aligned with the objective of visual recognition tasks. More details to be described in the ensuring subsections.

### 3.1 Historical Contrastive Instance Discrimination

The proposed HCID learns from unlabeled target samples via contrastive learning over their embeddings generated from current and historical models: the positive pairs are pulled close while negative pairs are pushed apart. It is a new type of contrastive learning for UMA, which preserves source-domain hypothesis by generating positive keys from historical models. HCID works at instance level and encourages to learn instance-discriminative target representations that generalize well to unseen data [98].

**HCID loss.** Given a query sample $x_q$ and a set of key samples $X_k = \{x_{k_0}, x_{k_1}, x_{k_2}, ..., x_{k_N}\}$, HCID employs current model $E^t$ to encode the query $q^t = E^t(x_q)$, and historical encoders $E^{t-m}$ to encode the keys $k_n^{t-m} = E^{t-m}(x_{k_n}), n = 0, \cdots, N$. With the encoded embeddings, HCID is achieved via a historical contrastive loss $\mathcal{L}_{\text{HisNCE}}$, minimization of which pulls $q$ close to its positive key $k_+^{t-m}$ while pushing it apart from all other (negative) keys:

$$\mathcal{L}_{\text{HisNCE}} = \sum_{x_q \in X_{tgt}} -\log \frac{\exp(q^t \cdot k_+^{t-m}/\tau) r_+^{t-m}}{\sum_{i=0}^{N} \exp(q^t \cdot k_i^{t-m}/\tau) r_i^{t-m}} \tag{1}$$

where $\tau$ is a temperature parameter [82], $r$ indicates the reliability of each key $k_n^{t-m}$, with which we reweight the similarity loss of each key to encourage to memorize well-learnt instead of poorly-learnt historical embeddings. In this work, we use the classification entropy to estimate the reliability of each key. The positive key sample is the augmentation of the query sample[22, 36], and all the rest are negative keys.

**Remark 1** *Note $\mathcal{L}_{HisNCE}$ in Eq.1 has a similar form as the InfoNCE loss[56, 22]. InfoNCE can actually be viewed as a special case of HisNCE, where all the query and keys are encoded by the current model ($m = 0$) and the reliability is fixed ($r_i^{t-m} = 1, \forall i$). For HisNCE, we assign each key a*

*reliability score to encourage to memorize the well-learnt historical embeddings only. It is also worth noting that Eq. 1 only shows historical contrast with one historical model for simplicity. In practice, we could employ multiple historical models to comprehensively distill (memorize) the well-learnt embeddings from them. It could be achieved by computing Eq.1 multiple times.*

### 3.2 Historical contrastive category discrimination

We design HCCD that generates pseudo labels and learns them conditioned on a historical consistency, *i.e.*, the prediction consistency across the current and historical models. HCCD can be viewed as a new type of self-training, where pseudo labels are re-weighted by the historical consistency. It works at category level and encourages to learn category-discriminative target representations that are aligned with the objective of visual recognition tasks in UMA.

**Historical contrastive pseudo label generation.** Given an unlabeled sample $x$, the current and historical models predict $p^t = G^t(x)$ (as the query) and $p^{t-m} = G^{t-m}(x)$ (as the keys). The pseudo label and the historical consistency of the sample are computed by:

$$
\hat{y} = \mathbf{\Gamma}(p^t),
$$
$$
h_{con} = 1 - \text{Sigmoid}(||p^t - p^{t-m}||_1),
$$

where $p$ is a $K$-class probability vector, $\mathbf{\Gamma}$ is the pseudo label generation function [103, 102] and $\hat{y} = (\hat{y}^{(1)}, \hat{y}^{(2)}, ..., \hat{y}^{(C)})$ is the predicted category label.

**HCCD loss.** Given the unlabeled data $x$ and its historical contrastive pseudo label $(\hat{y}, h_{con})$, HCCD performs self-training on target data $x$ via a weighted cross-entropy loss:

$$
\mathcal{L}_{\text{HisST}} = - \sum_{x \in X_{tgt}} h_{con} \times \hat{y} \log p_x, \tag{2}
$$

where $h_{con}$ is the per-sample historical consistency and we use it to re-weight the self-training loss. If the predictions of a sample across the current and historical models are consistent, we consider it as a well-learnt sample and increase its influence in self-training. Otherwise, we consider it as pooly-learnt sample and decrease its influence in self-training.

### 3.3 Theoretical Insights

The two designs in Historical Contrastive Learning (HCL) are inherently connected with some probabilistic models and convergent under certain conditions:

**Proposition 1** *The historical contrastive instance discrimination (HCID) can be modelled as a maximum likelihood problem optimized via Expectation Maximization.*

**Proposition 2** *The HCID is convergent under certain conditions.*

**Proposition 3** *The historical contrastive category discrimination (HCCD) can be modelled as a classification maximum likelihood problem optimized via Classification Expectation Maximization.*

**Proposition 4** *The HCCD is convergent under certain conditions.*

The proofs of **Proposition 1**, **Proposition 2**, **Proposition 3** and **Proposition 4** are provided in the appendix, respectively.

## 4   Experiments

This section presents experiments including datasets, implementation details, evaluations of the proposed HCL in semantic segmentation, object detection and image classification tasks as well as the discussion of its desirable features.

Table 2: Experiments on semantic segmentation task GTA5 → Cityscapes ("SF" denotes source-data free, *i.e.*, adaptation without source data).

| Method | SF | Road | SW | Build | Wall | Fence | Pole | TL | TS | Veg. | Terrain | Sky | PR | Rider | Car | Truck | Bus | Train | Motor | Bike | mIoU |
|---|---|---|---|---|---|---|---|---|---|---|---|---|---|---|---|---|---|---|---|---|---|
| AdaptSeg [76] | ✗ | 86.5 | 36.0 | 79.9 | 23.4 | 23.3 | 23.9 | 35.2 | 14.8 | 83.4 | 33.3 | 75.6 | 58.5 | 27.6 | 73.7 | 32.5 | 35.4 | 3.9 | 30.1 | 28.1 | 42.4 |
| AdvEnt [79] | ✗ | 89.4 | 33.1 | 81.0 | 26.6 | 26.8 | 27.2 | 33.5 | 24.7 | 83.9 | 36.7 | 78.8 | 58.7 | 30.5 | 84.8 | 38.5 | 44.5 | 1.7 | 31.6 | 32.4 | 45.5 |
| IDA [57] | ✗ | 90.6 | 37.1 | 82.6 | 30.1 | 19.1 | 29.5 | 32.4 | 20.6 | 85.7 | 40.5 | 79.7 | 58.7 | 31.1 | 86.3 | 31.5 | 48.3 | 0.0 | 30.2 | 35.8 | 46.3 |
| CRST [102] | ✗ | 91.0 | 55.4 | 80.0 | 33.7 | 21.4 | 37.3 | 32.9 | 24.5 | 85.0 | 34.1 | 80.8 | 57.7 | 24.6 | 84.1 | 27.8 | 30.1 | 26.9 | 26.0 | 42.3 | 47.1 |
| CrCDA [35] | ✗ | 92.4 | 55.3 | 82.3 | 31.2 | 29.1 | 32.5 | 33.2 | 35.6 | 83.5 | 34.8 | 84.2 | 58.9 | 32.2 | 84.7 | 40.6 | 46.1 | 2.1 | 31.1 | 32.7 | 48.6 |
| UR [72] | ✓ | 92.3 | 55.2 | 81.6 | 30.8 | 18.8 | 37.1 | 17.7 | 12.1 | 84.2 | 35.9 | 83.8 | 57.7 | 24.1 | 81.7 | 27.5 | 44.3 | 6.9 | 24.1 | 40.4 | 45.1 |
| +HCL | ✓ | 92.2 | 54.1 | 81.7 | 34.2 | 25.4 | 37.9 | 35.8 | 29.8 | 84.1 | 38.0 | 83.9 | 59.1 | 27.1 | 84.6 | 33.9 | 41.9 | 16.2 | 27.7 | 44.7 | 49.1 |
| SFDA [48] | ✓ | 91.7 | 52.7 | 82.2 | 28.7 | 20.3 | 36.5 | 30.6 | 23.6 | 81.7 | 35.6 | 84.8 | 59.5 | 22.6 | 83.4 | 29.6 | 32.4 | 11.8 | 23.8 | 39.6 | 45.8 |
| +HCL | ✓ | 92.3 | 54.5 | 82.6 | 33.1 | 26.2 | 38.9 | 37.9 | 31.7 | 83.1 | 38.1 | 84.4 | 60.9 | 30.0 | 84.5 | 32.6 | 41.2 | 14.2 | 26.4 | 43.2 | 49.3 |
| HCID | ✓ | 89.5 | 53 | 80.3 | 33.9 | 22.9 | 36.2 | 32.7 | 23.8 | 82.3 | 36.5 | 73.7 | 60.0 | 22.4 | 83.8 | 28.9 | 34.7 | 13.5 | 21.2 | 38.0 | 45.6 |
| HCCD | ✓ | 91.0 | 53.6 | 81.5 | 32.4 | 23.1 | 36.9 | 32.3 | 26.3 | 82.8 | 37.2 | 80.4 | 58.5 | 25.0 | 82.5 | 29.9 | 34.2 | 15.5 | 23.2 | 40.5 | 46.7 |
| HCL | ✓ | 92.0 | 55.0 | 80.4 | 33.5 | 24.6 | 37.1 | 35.1 | 28.8 | 83.0 | 37.6 | 82.3 | 59.4 | 27.6 | 83.6 | 32.3 | 36.6 | 14.1 | 28.7 | 43.0 | 48.1 |

Table 3: Experiments on semantic segmentation task SYNTHIA → Cityscapes ("SF" denotes source-data free, *i.e.*, adaptation without source data).

| Method | SF | Road | SW | Build | Wall$^*$ | Fence$^*$ | Pole$^*$ | TL | TS | Veg. | Sky | PR | Rider | Car | Bus | Motor | Bike | mIoU | mIoU$^*$ |
|---|---|---|---|---|---|---|---|---|---|---|---|---|---|---|---|---|---|---|---|
| AdaptSeg [76] | ✗ | 84.3 | 42.7 | 77.5 | - | - | - | 4.7 | 7.0 | 77.9 | 82.5 | 54.3 | 21.0 | 72.3 | 32.2 | 18.9 | 32.3 | - | 46.7 |
| AdvEnt [79] | ✗ | 85.6 | 42.2 | 79.7 | 8.7 | 0.4 | 25.9 | 5.4 | 8.1 | 80.4 | 84.1 | 57.9 | 23.8 | 73.3 | 36.4 | 14.2 | 33.0 | 41.2 | 48.0 |
| IDA [57] | ✗ | 84.3 | 37.7 | 79.5 | 5.3 | 0.4 | 24.9 | 9.2 | 8.4 | 80.0 | 84.1 | 57.2 | 23.0 | 78.0 | 38.1 | 20.3 | 36.5 | 41.7 | 48.9 |
| CRST [102] | ✗ | 67.7 | 32.2 | 73.9 | 10.7 | 1.6 | 37.4 | 22.2 | 31.2 | 80.8 | 80.5 | 60.8 | 29.1 | 82.8 | 25.0 | 19.4 | 45.3 | 43.8 | 50.1 |
| CrCDA[35] | ✗ | 86.2 | 44.9 | 79.5 | 8.3 | 0.7 | 27.8 | 9.4 | 11.8 | 78.6 | 86.5 | 57.2 | 26.1 | 76.8 | 39.9 | 21.5 | 32.1 | 42.9 | 50.0 |
| UR [72] | ✓ | 59.3 | 24.6 | 77.0 | 14.0 | 1.8 | 31.5 | 18.3 | 32.0 | 83.1 | 80.4 | 46.3 | 17.8 | 76.7 | 17.0 | 18.5 | 34.6 | 39.6 | 45.0 |
| +HCL | ✓ | 76.7 | 33.7 | 78.7 | 7.2 | 0.1 | 34.4 | 23.2 | 31.6 | 80.5 | 84.3 | 54.4 | 26.6 | 79.5 | 35.9 | 24.8 | 34.4 | 44.1 | 51.1 |
| SFDA [48] | ✓ | 67.8 | 31.9 | 77.1 | 8.3 | 1.1 | 35.9 | 21.2 | 26.7 | 79.8 | 79.4 | 58.8 | 27.3 | 80.4 | 25.3 | 19.5 | 37.4 | 42.4 | 48.7 |
| +HCL | ✓ | 78.2 | 35.3 | 79.6 | 7.3 | 0.2 | 37.7 | 21 | 30.9 | 80.4 | 83.3 | 59.8 | 29.4 | 79.2 | 34.2 | 24.5 | 38.9 | 45.0 | 51.9 |
| HCL | ✓ | 80.9 | 34.9 | 76.7 | 6.6 | 0.2 | 36.1 | 20.1 | 28.2 | 79.1 | 83.1 | 55.6 | 25.6 | 78.8 | 32.7 | 24.1 | 32.7 | 43.5 | 50.2 |

## 4.1 Datasets

**UMA for semantic segmentation** is evaluated on two challenging tasks GTA5 [61] → Cityscapes [15] and SYNTHIA [62] → Cityscapes. GTA5 has $24,966$ synthetic images and shares 19 categories with Cityscapes. SYNTHIA contains $9,400$ synthetic images and shares 16 categories with Cityscapes. Cityscapes has 2975 and 500 real-world images for training and validation, respectively.

**UMA for object detection** is evaluated on tasks Cityscapes → Foggy Cityscapes [68] and Cityscapes → BDD100k [89]. Foggy Cityscapes is derived by applying simulated fog to the $2,975$ Cityscapes images. BDD100k has $70k$ training images and $10k$ validation images, and shares 7 categories with Cityscapes. We evaluate a subset of BDD100k (*i.e.*, *daytime*) as in [84, 65, 13] for fair comparisons.

**UMA for image classification** is evaluated on benchmarks VisDA17 [58] and Office-31 [63]. VisDA17 has $152,409$ synthetic images as the source domain and $55,400$ real images of 12 shared categories as the target domain. Office-31 has 4110 images of three sources including 2817 from **A**mazon, 795 from **W**ebcam and 498 from **D**SLR (with 31 shared categories). Following [102, 63, 70], the evaluation is on six adaptation tasks A→W, D→W, W→D, A→D, D→A, and W→A.

## 4.2 Implementation Details

**Semantic segmentation:** Following [79, 103], we employ DeepLab-V2 [8] as the segmentation model. We adopt SGD [3] with momentum 0.9, weight decay $1e-4$ and learning rate $2.5e-4$, where the learning rate is decayed by a polynomial annealing policy [8].

**Object detection:** Following [84, 65, 13], we adopt Faster R-CNN [60] as the detection model. We use SGD [3] with momentum 0.9 and weight decay $5e-4$. The learning rate is $1e-3$ for first $50k$ iterations and then decreased to $1e-4$ for another $20k$ iterations.

**Image classification:** Following [102, 63, 70], we adopt ResNet-101 and ResNet-50 [23] as the classification models for VisDA17 and Office-31, respectively. We use SGD [3] with momentum 0.9, weight decay $5e-4$, learning rate $1e-3$ and batch size 32.

## 4.3 Unsupervised Domain Adaption for Semantic Segmentation

We evaluated the proposed HCL in UMA-based semantic segmentation tasks GTA5 → Cityscapes and SYNTHIA → Cityscapes. Tables 2 and 3 show experimental results in mean Intersection-over-Union (mIoU). We can see that HCL outperforms state-of-the-art UMA methods by large margins. In addition, HCL is complementary to existing UMA methods and incorporating it as

Table 4: Experiments on object detection task Cityscapes → Foggy Cityscapes ("SF" denotes source-data free, *i.e.*, adaptation without source data).

| Method | SF | person | rider | car | truck | bus | train | mcycle | bicycle | mAP |
|---|---|---|---|---|---|---|---|---|---|---|
| DA [13] | ✗ | 25.0 | 31.0 | 40.5 | 22.1 | 35.3 | 20.2 | 20.0 | 27.1 | 27.6 |
| MLDA [83] | ✗ | 33.2 | 44.2 | 44.8 | 28.2 | 41.8 | 28.7 | 30.5 | 36.5 | 36.0 |
| DMA [37] | ✗ | 30.8 | 40.5 | 44.3 | 27.2 | 38.4 | 34.5 | 28.4 | 32.2 | 34.6 |
| CAFA [27] | ✗ | 41.9 | 38.7 | 56.7 | 22.6 | 41.5 | 26.8 | 24.6 | 35.5 | 36.0 |
| SWDA [65] | ✗ | 36.2 | 35.3 | 43.5 | 30.0 | 29.9 | 42.3 | 32.6 | 24.5 | 34.3 |
| CRDA [84] | ✗ | 32.9 | 43.8 | 49.2 | 27.2 | 45.1 | 36.4 | 30.3 | 34.6 | 37.4 |
| SFOD [43] | ✓ | 25.5 | 44.5 | 40.7 | 33.2 | 22.2 | 28.4 | 34.1 | 39.0 | 33.5 |
| +HCL | ✓ | 39.3 | 46.7 | 48.6 | 32.9 | 46.2 | 38.2 | 33.9 | 36.9 | 40.3 |
| HCL | ✓ | 38.7 | 46.0 | 47.9 | 33.0 | 45.7 | 38.9 | 32.8 | 34.9 | 39.7 |

Table 5: Experiments on object detection task Cityscapes → BDD100k ("SF" denotes source-data free, *i.e.*, adaptation without source data).

| Method | SF | person | rider | car | truck | bus | mcycle | bicycle | mAP |
|---|---|---|---|---|---|---|---|---|---|
| DA [13] | ✗ | 29.4 | 26.5 | 44.6 | 14.3 | 16.8 | 15.8 | 20.6 | 24.0 |
| SWDA [65] | ✗ | 30.2 | 29.5 | 45.7 | 15.2 | 18.4 | 17.1 | 21.2 | 25.3 |
| CRDA [84] | ✗ | 31.4 | 31.3 | 46.3 | 19.5 | 18.9 | 17.3 | 23.8 | 26.9 |
| SFOD [43] | ✓ | 32.4 | 32.6 | 50.4 | 20.6 | 23.4 | 18.9 | 25.0 | 29.0 |
| +HCL | ✓ | 33.9 | 34.4 | 52.8 | 22.1 | 25.3 | 22.6 | 26.7 | 31.1 |
| HCL | ✓ | 32.7 | 33.2 | 52.0 | 21.3 | 25.6 | 21.5 | 26.0 | 30.3 |

denoted by "+HCL" improves the existing UMA methods clearly and consistently. Furthermore, HCL even achieves competitive performance as compared with state-of-the-art UDA methods (labeled by ✗in the column SF) which require to access the labeled source data in training. Further, We conduct ablation studies of the proposed HCL over the UMA-based semantic segmentation task GTA5 → Cityscapes. As the bottom of Table 2 shows, either HCID or HCCD achieves comparable performance. In addition, HCID and HCCD offer orthogonal self-supervision signals where HCID focuses on instance-level discrimination between queries and historical keys and HCCD focuses on category-level discrimination among samples with different pseudo category labels. The two designs are thus complementary and the combination of them in HCL produces the best segmentation.

## 4.4 Unsupervised Domain Adaptation for Object Detection

We evaluated the proposed HCL over the UMA-based object detection tasks Cityscapes → Foggy Cityscapes and Cityscapes → BDD100k. Tables 4 and 5 show experimental results. We can observe that HCL outperforms state-of-the-art UMA method SFOD clearly. Similar to the semantic segmentation experiments, HCL achieves competitive performance as compared with state-of-the-art UDA methods (labeled by ✗in column SF) which require to access labeled source data in training.

## 4.5 Unsupervised Domain Adaptation for Image Classification

We evaluate the proposed HCL over the UMA-based image classificat tasks VisDA17 and Office-31. Tables 6 and 7 show experimental results. We can observe that HCL outperforms state-of-the-art UMA methods clearly. Similar to the semantic segmentation and object detection experiments, HCL achieves competitive performance as compared with state-of-the-art UDA methods (labeled by ✗) which require to access labeled source data in training.

## 4.6 Discussion

**Generalization across computer vision tasks:** We study how HCL generalizes across computer vision tasks by evaluating it over three representative tasks on *semantic segmentation*, *object detection* and *image classification*. Experiments in Tables 2- 7 show that HCL achieves competitive performance consistently across all three visual tasks, demonstrating the generalization ability of HCL across computer vision tasks.

**Complementarity studies:** We study the complementarity of our proposed HCL by combining it with existing UMA methods. Experiments in Table 2 (the row highlighted by "+HCL") shows that incorporating HCL boosts the existing UMA methods consistently.

Table 6: Experiments on image classification benchmark VisDA17 ("SF" denotes source-data free, *i.e.*, adaptation without source data).

| Method | SF | Aero | Bike | Bus | Car | Horse | Knife | Motor | Person | Plant | Skateboard | Train | Truck | Mean |
|---|---|---|---|---|---|---|---|---|---|---|---|---|---|---|
| DANN [17] | ✗ | 81.9 | 77.7 | 82.8 | 44.3 | 81.2 | 29.5 | 65.1 | 28.6 | 51.9 | 54.6 | 82.8 | 7.8 | 57.4 |
| ENT [18] | ✗ | 80.3 | 75.5 | 75.8 | 48.3 | 77.9 | 27.3 | 69.7 | 40.2 | 46.5 | 46.6 | 79.3 | 16.0 | 57.0 |
| MCD [66] | ✗ | 87.0 | 60.9 | 83.7 | 64.0 | 88.9 | 79.6 | 84.7 | 76.9 | 88.6 | 40.3 | 83.0 | 25.8 | 71.9 |
| CBST [103] | ✗ | 87.2 | 78.8 | 56.5 | 55.4 | 85.1 | 79.2 | 83.8 | 77.7 | 82.8 | 88.8 | 69.0 | 72.0 | 76.4 |
| CRST [102] | ✗ | 88.0 | 79.2 | 61.0 | 60.0 | 87.5 | 81.4 | 86.3 | 78.8 | 85.6 | 86.6 | 73.9 | 68.8 | 78.1 |
| 3C-GAN [42] | ✓ | 94.8 | 73.4 | 68.8 | 74.8 | 93.1 | 95.4 | 88.6 | 84.7 | 89.1 | 84.7 | 83.5 | 48.1 | 81.6 |
| +HCL | ✓ | 93.8 | 86.6 | 84.1 | 74.3 | 93.2 | 95.0 | 88.4 | 85.0 | 90.4 | 85.2 | 84.5 | 49.8 | 84.2 |
| SHOT [45] | ✓ | 93.7 | 86.4 | 78.7 | 50.7 | 91.0 | 93.5 | 79.0 | 78.3 | 89.2 | 85.4 | 87.9 | 51.1 | 80.4 |
| +HCL | ✓ | 94.3 | 87.0 | 82.6 | 70.6 | 92.0 | 93.2 | 87.0 | 80.6 | 89.6 | 86.8 | 84.6 | 58.7 | 83.9 |
| HCL | ✓ | 93.3 | 85.4 | 80.7 | 68.5 | 91.0 | 88.1 | 86.0 | 78.6 | 86.6 | 88.8 | 80.0 | 74.7 | 83.5 |

Table 7: Experiments on image classification benchmark Office-31 ("SF" denotes source-data free, *i.e.*, adaptation without source data).

| Method | SF | A→W | D→W | W→D | A→D | D→A | W→A | Mean |
|---|---|---|---|---|---|---|---|---|
| DAN [49] | ✗ | 80.5 | 97.1 | 99.6 | 78.6 | 63.6 | 62.8 | 80.4 |
| DANN [17] | ✗ | 82.0 | 96.9 | 99.1 | 79.7 | 68.2 | 67.4 | 82.2 |
| ADDA [78] | ✗ | 86.2 | 96.2 | 98.4 | 77.8 | 69.5 | 68.9 | 82.9 |
| JAN [50] | ✗ | 85.4 | 97.4 | 99.8 | 84.7 | 68.6 | 70.0 | 84.3 |
| CBST [103] | ✗ | 87.8 | 98.5 | 100 | 86.5 | 71.2 | 70.9 | 85.8 |
| CRST [102] | ✗ | 89.4 | 98.9 | 100 | 88.7 | 72.6 | 70.9 | 86.8 |
| 3C-GAN [42] | ✓ | 93.7 | 98.5 | 99.8 | 92.7 | 75.3 | 77.8 | 89.6 |
| +HCL | ✓ | 93.4 | 99.3 | 100.0 | 94.6 | 77.1 | 79.0 | 90.6 |
| SHOT [45] | ✓ | 91.2 | 98.3 | 99.9 | 90.6 | 72.5 | 71.4 | 87.3 |
| +HCL | ✓ | 92.8 | 99.0 | 100.0 | 94.4 | 76.1 | 78.3 | 90.1 |
| HCL | ✓ | 92.5 | 98.2 | 100.0 | 94.7 | 75.9 | 77.7 | 89.8 |

**Feature visualization:** This paragraph presents the t-SNE [53] visualization of feature representation on GTA → Cityscapes model adaptation task. We compare HCL with two state-of-the-art UMA methods, *i.e.*, "UR" [72] and "SFDA" [48], and Fig.3 shows the visualization. We can observe that HCL can learn desirable instance-discriminative yet category-discriminative representations because it incorporates two key designs that work in a complementary manner: 1) HCID works at instance level, which encourages to learn instance-discriminative target representations that generalize well to unseen data [98]; 2) HCCD works at category level which encourages to learn category-discriminative target representations that are well aligned with the objective of down-stream visual tasks. In addition, qualitative illustrations are provided in Fig.4. It can be observed that our proposed HCL clearly outperforms UR and SFDA.

**Generalization across learning setups:** We study how HCL generalizes across learning setups by adapting it into two adaptation setups, *i.e.*, *partial-set adaptation* and *open-set adaptation*. Experiments in Table 8 show that HCL achieves competitive performance consistently across both setups.

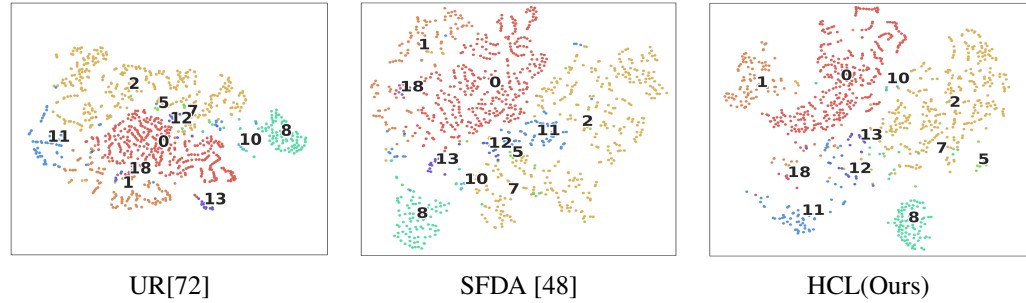

UR[72]  SFDA [48]  HCL(Ours)

Figure 3: The t-SNE [53] visualization of feature representation on GTA → Cityscapes unsupervsied model adaptation task: Each color in the graphs stands for a category of samples (image pixels) with a digit representing the center of a category of samples. It can be observed that the proposed HCL outperforms "UR" and "SFDA" qualitatively, by generating instance-discriminative and category-discriminative representations for unlabeled target data.

Table 8: Experiments on image classification benchmark Office-Home under the setup of partial-set DA (domain adaptation) and open-set DA ("SF" denotes source-data free, $i.e.$, adaptation without source data).

| Partial-set DA | SF | A→C | A→P | A→R | C→A | C→P | C→R | P→A | P→C | P→R | R→A | R→C | R→P | Mean |
|---|---|---|---|---|---|---|---|---|---|---|---|---|---|---|
| SAN [5] | ✗ | 44.4 | 68.7 | 74.6 | 67.5 | 65.0 | 77.8 | 59.8 | 44.7 | 80.1 | 72.2 | 50.2 | 78.7 | 65.3 |
| ETN [6] | ✗ | 59.2 | 77.0 | 79.5 | 62.9 | 65.7 | 75.0 | 68.3 | 55.4 | 84.4 | 75.7 | 57.7 | 84.5 | 70.5 |
| SAFN [85] | ✗ | 58.9 | 76.3 | 81.4 | 70.4 | 73.0 | 77.8 | 72.4 | 55.3 | 80.4 | 75.8 | 60.4 | 79.9 | 71.8 |
| SHOT [45] | ✓ | 57.9 | 83.6 | 88.8 | 72.4 | 74.0 | 79.0 | 76.1 | 60.6 | 90.1 | 81.9 | 68.3 | 88.5 | 76.8 |
| +HCL | ✓ | 66.9 | 85.5 | 92.5 | 78.3 | 77.2 | 87.1 | 78.3 | 65.1 | 90.7 | 82.4 | 68.7 | 88.4 | 80.1 |
| HCL | ✓ | 65.6 | 85.2 | 92.7 | 77.3 | 76.2 | 87.2 | 78.2 | 66.0 | 89.1 | 81.5 | 68.4 | 87.3 | 79.6 |
| Open-set DA | SF | A→C | A→P | A→R | C→A | C→P | C→R | P→A | P→C | P→R | R→A | R→C | R→P | Mean |
| OSBP [67] | ✗ | 56.7 | 51.5 | 49.2 | 67.5 | 65.5 | 74.0 | 62.5 | 64.8 | 69.3 | 80.6 | 74.7 | 71.5 | 65.7 |
| OpenMax [2] | ✗ | 56.5 | 52.9 | 53.7 | 69.1 | 64.8 | 74.5 | 64.1 | 64.0 | 71.2 | 80.3 | 73.0 | 76.9 | 66.7 |
| STA [47] | ✗ | 58.1 | 53.1 | 54.4 | 71.6 | 69.3 | 81.9 | 63.4 | 65.2 | 74.9 | 85.0 | 75.8 | 80.8 | 69.5 |
| SHOT [45] | ✓ | 62.5 | 77.8 | 83.9 | 60.9 | 73.4 | 79.4 | 64.7 | 58.7 | 83.1 | 69.1 | 62.0 | 82.1 | 71.5 |
| +HCL | ✓ | 64.2 | 78.3 | 83.0 | 61.1 | 72.2 | 79.6 | 65.5 | 59.3 | 80.6 | 80.1 | 72.0 | 82.8 | 73.2 |
| HCL | ✓ | 64.0 | 78.6 | 82.4 | 64.5 | 73.1 | 80.1 | 64.8 | 59.8 | 75.3 | 78.1 | 69.3 | 81.5 | 72.6 |

| UR[72] | SFDA[48] | HCL(Ours) | Ground Truth |
|---|---|---|---|

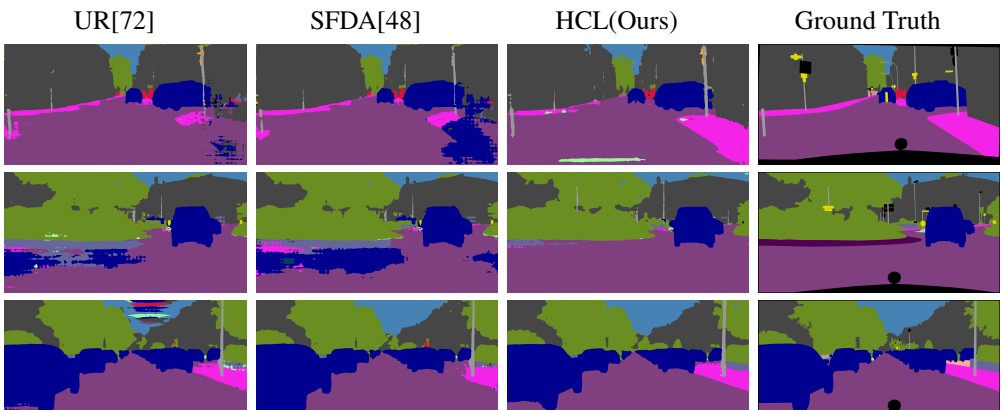

Figure 4: Qualitative illustrations and comparison over domain adaptive semantic segmentation task GTA5 → Cityscapes. Our historical contrastive learning (HCL) exploits historical source hypothesis to make up for the absence of source data in UMA, which produces better qualitative results ($i.e.$, semantic segmentation) by preserving the source hypothesis. It can be observed that HCL generates better segmentation results, for example, the sidewalk in the first row, the road in the second row and the sky and sidewalk in the third row.

## 5 Conclusion

In this work, we studied historical contrastive learning, an innovative UMA technique that exploits historical source hypothesis to make up for the absence of source data in UMA. We achieve historical contrastive learning by novel designs of historical contrastive instance discrimination and historical contrastive category discrimination which learn discriminative representations for target data while preserving source hypothesis simultaneously. Extensive experiments over a variety of visual tasks and learning setups show that HCL outperforms state-of-the-art techniques consistently. Moving forward, we will explore memory-based learning in other transfer learning tasks.

## Acknowledgement

This research was conducted at Singtel Cognitive and Artificial Intelligence Lab for Enterprises (SCALE@NTU), which is a collaboration between Singapore Telecommunications Limited (Singtel) and Nanyang Technological University (NTU) that is supported by A*STAR under its Industry Alignment Fund (LOA Award number: I1701E0013).

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
