# Model Adaptation: Historical Contrastive Learning for Unsupervised Domain Adaptation without Source Data

# Supplemental Materials

## A  Theoretical insights of HCL

### A.1  Proof of Proposition 1

**Proposition 1** *The historical contrastive instance discrimination (HCID) can be modelled as a maximum likelihood problem optimized via Expectation Maximization.*

*Proof:*

Maximum likelihood (ML) is a concept to describe the theoretic insights of clustering algorithms. For unsupervised model adaptation task, the objective of HCID is to adapt the source-trained encoder weights $\theta_E$ to maximize the log-likelihood function of the unlabeled target data $X_{tgt}$:

$$\theta_E^* = \arg\max_{\theta_E} \sum_{x_q \in X_{tgt}} \log p(x_q; \theta_E). \tag{1}$$

We assume that the unlabeled samples $X_{tgt}$ are related to latent variable $\{k_n\}_{n=1}^N$ which denotes the keys of the data and $N$ is the number of keys. In this way, we can re-write Eq. 1 as the following:

$$\theta_E^* = \arg\max_{\theta_E} \sum_{x_q \in X_{tgt}} \log \sum_{n=1}^N p(x_q, k_n; \theta_E) \tag{2}$$

As it is not easy to optimize Eq.14 directly, we employ a surrogate function to lower-bound the log-likelihood function:

$$
\begin{aligned}
\sum_{x_q \in X_{tgt}} \log \sum_{n=1}^N p(x_q, k_n; \theta_E) &= \sum_{x_q \in X_{tgt}} \log \sum_{n=1}^N \mathcal{Z}(k_n) \frac{p(x_q, k_n; \theta_E)}{\mathcal{Z}(k_c)} \\
&\geq \sum_{x_q \in X_{tgt}} \sum_{n=1}^N \mathcal{Z}(k_n) \log \frac{p(x_q, k_n; \theta_E)}{\mathcal{Z}(k_n)},
\end{aligned}
\tag{3}
$$

where $\mathcal{Z}(k_n)$ denotes some distribution over $k$'s ($\sum_{n=1}^N \mathcal{Z}(k_n) = 1$), and the last step of derivation employs Jensen's inequality [6, 7, 4]. This equality holds if $\frac{p(x_t, k_c; \theta_{f_q})}{\mathcal{D}(k_c)} =$ Constant, based on which we can get:

$$\mathcal{Z}(k_n) = \frac{p(x_q, k_n; \theta_E)}{\sum_{n=1}^N p(x_q, k_n; \theta_E)} = \frac{p(x_q, k_n; \theta_E)}{p(x_q; \theta_E)} = p(k_n; x_q, \theta_E) \tag{4}$$

Submitted to 35th Conference on Neural Information Processing Systems (NeurIPS 2021). Do not distribute.

By ignoring the constant $-\sum_{x_q \in X_{tgt}} \sum_{n=1}^{N} \mathcal{Z}(k_n) \log \mathcal{Z}(k_n)$ in Eq.3, we are supposed to maximize:

$$\sum_{x_q \in X_{tgt}} \sum_{n=1}^{N} \mathcal{Z}(k_n) \log p(x_q, k_n; \theta_E) \tag{5}$$

**Expectation step** focuses on estimating the posterior probability $p(k_n; x_q, \theta_E)$. We first generate keys by a historical encoder: $k_n^{t-m} = E^{t-m}(x_t)$, and $x_t \in X_{tgt}$. Then, We calculate $p(k_n; x_q, \theta_E) = p(k_n^{t-m}; x_q, \theta_E) = \mathbb{1}(x_q, k_n^{t-m})$, where $\mathbb{1}(x_q, k_n^{t-m}) = 1$ if both belong to the positive pair; otherwise, $\mathbb{1}(x_q, k_n^{t-m}) = 0$.

Please note the notation "$t - m$" shows that the $k$ is encoded by a historical encoder.

**Maximization step** focuses on maximizing the lower-bound in Eq.5. With the result from Expectation step, we get:

$$\sum_{x_q \in X_t} \sum_{n=1}^{N} \mathcal{Z}(k_n) \log p(x_q, k_n; \theta_E) = \sum_{x_q \in X_t} \sum_{n=1}^{N} p(k_n; x_q, \theta_E) \log p(x_q, k_n; \theta_E)$$

$$= \sum_{x_q \in X_t} \sum_{n=1}^{N} p(k_n^{t-m}; x_q, \theta_E) \log p(x_q, k_n^{t-m}; \theta_E) \tag{6}$$

$$= \sum_{x_q \in X_t} \sum_{n=1}^{N} \mathbb{1}(x_q, k_n^{t-m}) \log p(x_q, k_n^{t-m}; \theta_E)$$

By assuming a uniform prior over categorical keys, we have:

$$p(x_q, k_n^{t-m}; \theta_E) = p(x_q; k_n^{t-m}, \theta_E) p(k_n^{t-m}; \theta_E) = \frac{1}{N} \cdot p(x_q; k_n^{t-m}, \theta_E), \tag{7}$$

where we let the prior probability $p(k_n^{t-m}; \theta_E)$ for each $k_n$ as $1/N$ as all samples are evenly sampled as keys.

By assuming that the feature distribution around each key $k_n^{t-m}$ is an isotropic Gaussian [2], we have:

$$p(x_q; k_n^{t-m}, \theta_E) = \exp\left(\frac{-(q - k_+^{t-m})^2}{2\sigma_+^2}\right) \Big/ \sum_{n=1}^{N} \exp\left(\frac{-(q - k_n^{t-m})^2}{2\sigma_n^2}\right), \tag{8}$$

where $q = E(x_q)$, and we define $k_+^{t-m}$ as the positive key that is encoded by a historical encoder. By applying $\ell_2$-normalization on $q$ and $k$, we have $(q - k)^2 = 2 - 2q \cdot k$. Combining this equation with Eqs.14, 3, 5, 6, 7, 8, we re-write the likelihood maximization as:

$$\theta_E^* = \arg\min_{\theta_E} \sum_{x_q \in X_{tgt}} -\log \frac{\exp(q \cdot k_+^{t-m}/\tau_+)}{\sum_{n=1}^{N} \exp(q \cdot k_n^{t-m}/\tau_n)}, \tag{9}$$

where $\tau \propto \sigma^2$ stands for the density of the feature distribution around a key ($e.g.$, $k_n^{t-m}$).

In practice, we achieve Eq. 9 by minimizing a historical contrastive instance discrimination loss:

$$\mathcal{L}_{\text{HisNCE}} = \sum_{x_q \in X_{tgt}} -\log \frac{\exp(q^t \cdot k_+^{t-m}/\tau) r_+^{t-m}}{\sum_{i=0}^{N} \exp(q^t \cdot k_i^{t-m}/\tau) r_i^{t-m}} \tag{10}$$

Please note that Eq. 10 is an instance of Eq. 9. The two equations look different due to: 1) Eq. 10 adds the notation $t$ on $q$ to show that the $q$ is encoded by current encoder $E$ ($i.e.$, $\theta_E$). 2) Eq. 10 adds reliability $r$ to re-weight the loss for better implementation.

## A.2   Proof of Proposition 2

**Proposition 2** *The HCID is convergent under certain conditions.*

 *Proof:*

 We suppose

$$Q(\theta_E) = \sum_{x_q \in X_{tgt}} \log p(x_q; \theta_E) = \sum_{x_q \in X_{tgt}} \log \sum_{n=1}^{N} p(x_q, k_n; \theta_E)$$

$$= \sum_{x_q \in X_{tgt}} \log \sum_{n=1}^{N} \mathcal{Z}(k_n) \frac{p(x_q, k_n; \theta_E)}{\mathcal{Z}(k_n)} \quad (11)$$

$$\geq \sum_{x_q \in X_{tgt}} \sum_{n=1}^{N} \mathcal{Z}(k_n) \log \frac{p(x_q, k_n; \theta_E)}{\mathcal{Z}(k_n)}.$$

We have illustrated in Section A.1 that the inequality in Eq.11 holds with equality if $\mathcal{Z}(k_n) = p(k_n; x_q, \theta_E)$.

In the $i$-th Expectation-step, we have $\mathcal{Z}^i(k_n) = \mathcal{Z}^i(k_n^{t-m}) = p(k_n^{t-m}; x_q, \theta_E^i)$. As a result, we can have:

$$Q(\theta_E^i) = \sum_{x_q \in X_{tgt}} \sum_{n=1}^{N} \mathcal{Z}^i(k_n^{t-m}) \log \frac{p(x_q, k_n^{t-m}; \theta_E^i)}{\mathcal{Z}^i(k_n^{t-m})}. \quad (12)$$

In the $i$-th Maximization-step, $\mathcal{Z}^i(k_n^{t-m}) = p(k_n^{t-m}; x_q, \theta_E^i)$ is fixed, and the weights $\theta_E$ is optimized to maximize Equation 12. In this way, we can always have:

$$Q(\theta_E^{i+1}) \geq \sum_{x_q \in X_{tgt}} \sum_{n=1}^{N} \mathcal{Z}^i(k_n^{t-m}) \log \frac{p(x_q, k_n^{t-m}; \theta_E^{i+1})}{\mathcal{Z}^i(k_n^{t-m})}$$

$$\geq \sum_{x_q \in X_{tgt}} \sum_{n=1}^{N} \mathcal{Z}^i(k_n^{t-m}) \log \frac{p(x_q, k_n^{t-m}; \theta_E^i)}{\mathcal{Z}^i(k_n^{t-m})} \quad (13)$$

$$= Q(\theta_E^i).$$

Eq. 13 shows that $Q(\theta_E^i)$ monotonously increase along with Expectation-Maximization iterations.

As the log-likelihood is upper-bounded, *i.e.*, $Q(\theta_E^i) \leq 0$, the proposed historical contrastive instance discrimination will converge.

One possible way to achieve Eq. 13 is to conduct gradient descent by minimizing the historical contrastive instance discrimination loss in Eq. 10. Under a proper learning rate, this loss is guaranteed to decrease monotonically. In practical scenarios, model training is conventionally implemented via mini-batch gradient descent instead of gradient descent. This training strategy cannot strictly guarantee the monotonic decrease of the loss, but is supposed to converge to a lower one certainly.

## A.3 Proof of Proposition 3

**Proposition 3** *The historical contrastive category discrimination (HCCD) can be modelled as a classification maximum likelihood problem optimized via Classification Expectation Maximization.*

*Proof:*

Classification Maximum likelihood (CML) has been utilized to describe the theoretic insights of semi-supervised learning algorithms [1], and can be optimized via Classification Expectation Maximization (CEM). Different from the classical expectation maximization (mentioned in Section A.1) that consists of "expectation" and maximization steps, CEM involves an extra "classification" step (between them) that classifies a sample into a category with the maximum posterior probability [1, 9].

In [1], CML is formulated for the learning setup that includes both labeled and unlabeled data, which is defined as:

$$\theta_G^* = \arg\max_{\theta_G} \sum_{x_s \in X_{src}} \sum_{k=1}^{K} \hat{y}_s^{(k)} \log p(k; x_s, \theta_G) + \sum_{x_t \in X_{tgt}} \sum_{k=1}^{K} \hat{y}_t^{(k)} \log p(k; x_t, \theta_G). \tag{14}$$

For unsupervised model adaptation task, the objective of HCCD is to adapt the source-trained model weights $\theta_G$ to maximize the classification likelihood function of the unlabeled target data $X_{tgt}$. By removing the first term of the right-hand side (RHS) in Eq. 14, we get:

$$\theta_G^* = \arg\max_{\theta_G} \sum_{x_t \in X_{tgt}} \sum_{k=1}^{K} \hat{y}_t^{(k)} \log p(k; x_t, \theta_G). \tag{15}$$

Next, we can re-write HCCD as the weighted classification maximum likelihood:

$$\begin{aligned} \arg\min_{\theta_G} \mathcal{L}_{\text{HisST}} &= \arg\min_{\theta_G} - \sum_{x_t \in X_{tgt}} h_{con} \times \hat{y} \log p_{x_t} \\ &= \arg\max_{\theta_G} \sum_{x \in X_{tgt}} h_{con} \sum_{k=1}^{K} \hat{y}_t^{(k)} \log p(k; x_t, \theta_G), \end{aligned} \tag{16}$$

It can be observed that Eq. 16 is the same as Eq. 15 except involving an extra weighting element $h_{con} = 1 - \text{Sigmoid}(||p^t - p^{t-m}||_1)$.

In the following, we show the optimization of Eq. 16 is a CEM process.

**Expectation-step:** We estimate $p(k; x_t, \theta_G)$ for all $x_t \in X_{tgt}$.

**Classification-step:** We get $\hat{y}$ and $h_{con}$ for all $x_t \in X_{tgt}$, as follows:

$$\hat{y}^* = \arg\max_{\hat{y}} \sum_{k=1}^{K} \hat{y}_{(k)} \log p(k; x_t, \theta_G^t), \; s.t. \; \hat{y} \in \Delta^K, \tag{17}$$

$$h_{con} = 1 - \text{Sigmoid}(\sum_{k=1}^{K} p(k; x_t, \theta_G^t) - p(k; x_t, \theta_G^{t-m}))), \tag{18}$$

**Maximization-step:** With calculated $\hat{y}$ and $h_{con}$, we optimize $\theta_G$ as follows:

$$\arg\min_{\theta_G} - \sum_{x_t \in X_{tgt}} h_{con} \times \hat{y} \log p_{x_t}. \tag{19}$$

## A.4 Proof of Proposition 4

**Proposition 4** *The HCCD is convergent under certain conditions.*

*Proof:* We can re-arrange the three steps mentioned in previous subsection into two steps: 1) Expectation-classification step, and 2) Maximization step. Eq. 17 in the Expectation-classification step is a concave problem which has a globally optimal solution. The Maximization step is supervised learning, which is normally convergent [8, 5, 3]. Thus, the overall training process of HCCD is convergent.