# OpenReview forum: "Model Adaptation: Historical Contrastive Learning for Unsupervised Domain Adaptation without Source Data"
_NeurIPS.cc/2021/Conference — NeurIPS 2021 Poster_

### Official Review · Reviewer_UV3u · 2021-07-15

**Rating:** 6
**Confidence:** 4

**Summary:**

This paper studies the unsupervised model adaptation problem, where only the trained model, but not the source data, is available to use for target domain adaptation. The authors propose historical contrastive learning (HCL) technique to help this source-free setting. HCL is composed of two components, HCID and HCCD. HCID learns from target instances by contrasting the features generated from the adapted model and historical models. HCCD explores self-training to pseudo label target instances to learn category discriminative representations. Experiments on three different vision applications are conducted. The paper is generally easy to follow, despite some unclear points. The major concern is the novelty.

**Limitations And Societal Impact:**

No known concern on limitations and potential negative societal impact.

**Main Review:**

Strong points
1.	Model adaptation without access to source data is an interesting and practical problem.
2.	The experiments include three computer vision applications, segmentation, detection and classification. The performances on different datasets are consistently better than baselines, although some improvements are marginal (see week points for detail)

Weak points
1.	Limited novelty is my major concern. I recognize the authors apply contrastive learning to this new problem, and I also recognize the proposed modifications, e.g. the reliability scaler in Eq 3 compared to the traditional InfoNCE. But these do not meet the NeruIPS bar, especially the proposed modifications are not fully justified by experiments. (see below)
2.	For $r$, to support the claim that this soft version (compared to traditional $r_i^{t-m} =1$), the analytical experiment of $r=1$ is missing. Similarly, in Eq 4, a baseline by setting $h_{con} =1$ (no reweighting) is missing.
3.	Missing key related references [ref-1].
[ref-1]Vinod K Kurmi, Venkatesh K Subramanian, and Vinay P Namboodiri. Domain impression: A source data free domain adaptation method. WACV 2021
[ref-2] Jogendra Nath Kundu, et al., Towards Inheritable Models for Open-Set Domain Adaptation, CVPR 2020
Please consider making a discussion/comparison.
4.	The equation above line 164, $p^t$ is a scalar, l-1 norm on a scalar does not seem correct to me. One more right bracket.
5.	A completed ablation study on different applications is highly recommended.

Other minors:
6.     For the baselines, I suggest removing all the non-source-free ones, which are out of this paper’s focus.
7.	Line 121, the reference [20] seems wrong.


**Time Spent Reviewing:**

4

---

> ### Author Response · Authors · 2021-08-10
> **Thank you for your confirmation of the value of our studied research problem as well as the clearness in presentation.**
>
> Thank you for your confirmation of the value of our studied research problem as well as the clearness in presentation. Below please find our responses regarding your concerns.
>
> Q1: The novelty is limited?
> * We would clarify that the proposed historical contrastive instance discrimination (HCID) is a new type of contrastive learning for model adaptation which consists of two novel designs. The major one is to generate keys from historical models for preserving source-domain hypothesis. The minor one is to assign each key a reliability score for encouraging to memorize well-learnt historical embeddings only. Extensive experiments demonstrate the clear effectiveness of these two designs (partial results shown in the table below).
> * In addition, most existing contrastive learning studies [21,9,70,10] use a very similar NCE loss. The major differences among them are dictionary creation (or key generation) strategies: [21,10] generate keys by a momentum-updated encoder; [70] generates keys every epoch and stores them in a memory bank; [9] generates keys using samples from the current minibatch. Differently, our HCID generates keys from historical models for preserving source-domain hypothesis in source-free domain adaptation. Hence, our contribution is far beyond just adding a reliability score.
> * We experimentally verified the effectiveness of the proposed designs by comparing them (HCID, HCCD and HCL) with three representative contrastive learning methods: MoCo [21] (Kaiming etc.) SimCLR [9] (Hinton etc.) and InstDisc [70] (Zhirong etc.) in Table 2 of the appendix. We combine it with the Table 1 of the manuscript as shown below. It can be observed that our methods work well for source-free domain adaptation while the three representative contrastive learning methods (designed for unsupervised representation learning) are far less effective. Please refer to Section B.2 (Line.106-114) of the appendix for detailed analysis of these results.
>
> Table 1: Comparison with existing instance contrastive learning methods on UMA-based semantic segmentation task GTA-to-Cityscapes.
>
> | Method                          | mIoU    | gain  |
> | --------------------------------|:-------:|:-----:|
> | Baseline                        | 36.6    | N.A.  |
> | SimCLR [9] (Hinton etc.)        | 38.4    | +1.8  |
> | InstDisc [70] (Zhirong etc.)    | 38.0    | +1.4  |
> | MoCo [21] (Kaiming etc.)        | 38.9    | +2.3  |
> | HCID w/o $r$                    | 44.2    | +7.6  |
> | HCID w/ $r$                     | 45.6    | +9.0  |
> | HCCD w/o $h_{con}$                    | 42.6    | +6.0  |
> | HCCD w/ $h_{con}$                     | 46.7    | +10.1 |
> | HCL (HCID w/ $r$ + HCCD w/ $h_{con}$)| **48.1**| **+11.5**|
>
> Q2: The performance of HCID without $r$ and HCCD without $h_{con}$?
> * Thank you for the suggestion! We conducted the suggested experiments (i.e. the HCID without $r$ and HCCD without $h_{con}$ in the above table). It can be seen that HCID w/o $r$ outperforms the recent contrastive learning methods by over 5\% in mIoU, which is largely attributed to our historical contrast design. Including the reliability score (in HCID w/ $r$) further improves the mIoU by 1.4\%. In addition, HCCD w/o $h_{con}$ becomes a basic self-training method and further including our proposed historical consistency (HCCD w/ $h_{con}$) improves mIoU by 4.1\%. Thank you for your suggestion and we will include the two case studies into the updated manuscript.
>
> Q3: Missing related works [ref-1] Vinod K Kurmi (WACV'2021) and [ref-2] Jogendra (CVPR'2020)?
> * Thank you for sharing the two works and we will review them in the updated manuscript. We note that [ref-1] tackles source-free DA in a generative manner ($e.g.$, by generating samples from the source classes) whereas our HCL addresses it from the perspective of contrastive learning. [ref-2] focuses on the open-set DA problem whereas we focus on close-set DA. In addition, HCL outperforms [ref-1] and [ref-2] by large margins. For [ref-1], HCL outperforms it by 6.3\% in mean accuracy over close-set DA (Office-31 benchmark). For [ref-2], HCL outperforms it by 3.0\% in mean accuracy over open-set DA (Office-Home benchmark).
>
> Q4: $p^{t}$ is a scalar?
> * As described in Line 164, $p^{t}$ is a vector. We will clarify it further in the revised manuscript.
>
> Q5: More ablation study?
> * Thank you for the suggestion. We will include more ablation studies on different tasks and datasets in the revised manuscript.
>
> Q6: Removing non-source-free methods?
> * Thank you for your suggestion. As in previous studies [37,61,33,35], we include comparisons with non-source-free DA methods with two purposes: 1) to use them as references to show how source-free methods benchmark with non-source-free methods; 2) there are limited source-free studies for comparisons. Nevertheless, we will consider removing non-source-free methods for simplicity.
>
> Q7: The reference [20] seems wrong (in Line 121)?
> * The reference [20] (Dimensionality reduction by learning an invariant mapping) has no problem. Note [20] is the very first work that considers contrastive learning in machine learning which learns mappings that are invariant to certain transformations of the input. MoCo [21] also cited [20] as the very first work of contrastive learning at the beginning of Section 3.1.

---

> > ### Comment · Reviewer_UV3u · 2021-08-24
> > **Thank you for the detailed reponse.**
> >
> > Thanks for the detailed response. My concerns have been addressed and I upgrade my rating to 6.

---

### Official Review · Reviewer_8f8p · 2021-07-15

**Rating:** 7
**Confidence:** 4

**Summary:**

This paper proposes a generic method HCL to solve the source data-free domain adaptation problem across various computer vision tasks. Based on the memory mechanism, the method involves two designs to learn discriminative features while preserving the source hypothesis. The first design HCID leverages the contrastive instance discrimination where positive pairs are made of the feature from the current model and features from historical models including the original source model and model in previous epochs. The second design HCCD leverages the consistency between the current model and previous models to select confident pseudo labels for task-specific self-training. Experiments across various vision tasks and domain adaptation settings validate the effectiveness of both designs and the generic method.

**Limitations And Societal Impact:**

Yes.

**Main Review:**

## Pros
1. Originality: The HCID design and the HCL method are novel for source data-free domain adaptation.
2. Quality: The proposed method is technically sound and supported by theoretical analysis.
3. Clarity: The writing is clear and easy to follow. The paper is well-organized.
4. Significance: This paper targets the generic unsupervised model adaptation (UMA) problem, which is significant. Demonstratable results are achieved by HCL on various UMA benchmarks.

## Cons
1. Originality: Missing related works [A, B] on the recent popular UMA problem, although some results in these works may be better than the results in this paper.
2. Clarity: How HCL works with the two designs (HCID and HCCD) is unclear. $L_{HCL} = L_{HisNCE} + L_{HisST}$? Is there any tradeoff hyper-parameter between two objectives or do the two designs work in the meantime or in a two-stage fashion? If there is a tradeoff, how does the tradeoff hyper-parameter affect the adaptation performance? If there are other hyper-parameters, please clarify their effects similarly. In addition, the implementation or training details of the HCL applied to various computer vision tasks or domain adaptation settings are not provided adequately.

References

[A] Kundu et al., Universal source-free domain adaptation, CVPR'20

[B] Liang et al., Source Data-absent Unsupervised Domain Adaptation through Hypothesis Transfer and Labeling Transfer, arxiv'20

-----------------------------------------------------------------------------------
## Post-rebuttal
The response has addressed most of my concerns on the training details of the proposed method. I decided to keep my original score of 7 because of the strong performance and extensive experiments in the submission.


**Time Spent Reviewing:**

5h

---

> ### Author Response · Authors · 2021-08-10
> **Thank you for your acknowledgment of the novelty of our proposed methods, technical soundness with theoretical analysis, and impressive experimental results on various UMA benchmarks.**
>
> Thank you for your acknowledgment of the novelty of our proposed methods, technical soundness with theoretical analysis, and impressive experimental results on various UMA benchmarks. Below please find our responses regarding your concerns.
>
> Q1: Refer to [A] Kundu et al. (CVPR'2020) and [B] Liang et al. (arxiv'2020)?
> * Thank you for sharing these two works. We will review them in the updated paper. We note that [A] focuses on universal domain adaptation while we study closed-set domain adaptation. The two studies have different baselines and task settings as well as incomparable experimentation. We did not compare with [B] as it was not published.
>
> Q2: How HCL works with HCID and HCCD? $L_{HCL} = L_{HisNCE} + L_{HisST}$? Is there any other other hyper-parameter?
> * Yes, the overall training loss in HCL is $L_{HCL} = L_{HisNCE} + L_{HisST}$, i.e. the two objectives are combined with equal weight across all evaluated tasks and datasets. Our purpose is to avoid an extra weight parameter which often involves a sophisticated and time-consuming fine-tuning process for specific tasks/datasets. Nevertheless, we do know that fine-tuning the weight parameter often improves the domain adaptation performance.
> *Our HCL introduces no extra hyper-parameters. Note that for fair comparisons, we adopt those basic training hyper-parameters ($e.g.$, learning rate, batch size, temperature $\tau$, etc.) as used in most prior studies [65,68,86,54,70,9].
>
> Q3: More implementation details of the HCL?
> * We would clarify that we strictly followed the implementation details of prior studies [68,86,72,35] for all evaluated tasks and datasets. We only provide key implementation details due to the space limit. We will include more implementation details in the appendix later.
> * Besides, we are committed to open-source research and will release our implementation codes (including all training details) upon the acceptance of this work.

---

### Official Review · Reviewer_acZE · 2021-07-16

**Rating:** 8
**Confidence:** 5

**Summary:**

The authors address unsupervised model adaptation (UMA). While unsupervised domain adaptation (UDA) lets the model access the source domain data during an adaptation to a target domain, such access could raise multiple concerns such as privacy and portability. UMA does not allow the model to access the source domain data during adaptation. The proposed method, historical contrastive learning (HCL), utilizes the historical source hypothesis to remedy the lack of source data. The experimental results show that the proposed method achieves state-of-the-art performance for multiple visual recognition tasks.

**Ethical Concerns:**

There are no ethical concerns.


**Limitations And Societal Impact:**

In the checklists, there are descriptions of the limitations and the societal impacts. The societal impact here is a positive one, but the reviewer agrees with the authors.

**Main Review:**

Originality:
- UMA has two original components; one is historical contrastive instance discrimination (HCID), and the other is historical contrastive category discrimination (HCCD).
- The idea of contrastive learning between two models can be found in another paper for continual learning [a]. It is not a problem since [a] is released after the deadline of NeurIPS 2021.

[a] Cha et al., Co2L: Contrastive Continual Learning. arXiv, 2021.

Quality:
- While there are already many references, the authors should refer to the following papers.
- [b] addresses source-free domain adaptation under universal domain adaptation settings, which means there is no guarantee that the classes in the target domain are the same as in the source domain. Since UMA in this paper is a variant of closed domain adaptation, [b] can be considered a more complex problem.
- [c] and [d] address source free domain adaptation for image classification. [c] does not compare the performance to those of existing UDA methods. [d] reports some results on VisDA, Office-31, and Office-Home. [d] uses AlexNet for Office-31; therefore, the performances are worse than those in Table 7.
- By the way, [37] is already published in CVPR 2021.

[b] Kundu et al., Universal Source-Free Domain Adaptation. CVPR, 2020.

[c] Kurmi et al., Domain Impression: A Source Data Free Domain Adaptation Method. WACV, 2021.

[d] Yeh et al., SoFA: Source-data-free Feature Alignment for Unsupervised Domain Adaptation. WACV, 2021.

Clarity:
- The motivation and formulation are clearly described.
- The theoretical insights in Section 3.5 clarifies the use of the proposed method for UMA.

Significance:
- This paper's major strength is throughout experiments for various visual recognition tasks such as segmentation, detection, and classification. HCL itself shows the state-of-the-art performance; moreover, the combination with other state-of-the-art methods shows that HCL is complementary to each.

Post-rebuttal:
- I have read the other reviews and all responses from the authors. There are no concerns to recommend this paper for acceptance. Therefore, I would like to keep my first score.

**Time Spent Reviewing:**

2 hours

---

> ### Author Response · Authors · 2021-08-10
> **Thank you for your appreciation of the motivation and formulation of this research, the technical method together with the theoretical insights as well as comprehensive experiments.**
>
> Thank you for your appreciation of the motivation and formulation of this research, the technical method together with the theoretical insights as well as comprehensive experiments. Below please find our response regarding your concerns.
>
> Q1: Refer to [a] Cha et al., Co2L: Contrastive Continual Learning. arXiv, 2021 ?
> * Thank you for sharing this work. We didn’t review [a] as it is released after the NeurIPS deadline. We will review it in the updated paper. We note that [a] considers supervised continual learning from previously learned tasks to a new task, and it explores contrastive learning to learn and preserve representations continually along the supervised learning of new task. Our HCL instead considers unsupervised model adaptation across domains, and it explores contrastive learning to preserve and adapt source-hypothesis in an unsupervised manner. The two studies thus have clearly different motivations and designs though both employ contrastive learning.
>
> Q2: Refer [b] Kundu et al. (CVPR'2020), [c] Kurmi et al. (WACV2021), [d] Yeh et al. (WACV2021)?
> * Thank you for sharing these three studies and we will review them in the updated paper. We agree that [b] addresses a universal domain adaptation problem whereas we focus on close-set DA. In addition, we tackle source-free domain adaptation from the perspective of contrastive learning while [c] and [d] tackle this problem in a generative manner by generating samples from the source classes and generating reference distributions, respectively.

---

### Official Review · Reviewer_DhdD · 2021-07-19

**Rating:** 7
**Confidence:** 4

**Summary:**

In this paper, the authors propose to solve the problem of unsupervised model adaptation. In this task, the aim is that given a classifier trained on a source dataset and a set of unlabelled target examples, an unsupervised model adaptation is performed to improve the performance of the model on the target dataset without requiring access to source examples. The method is evaluated on standard domain adaptation datasets for classification, detection and segmentation. The approach is related to EM and some theoretical insights provided.


**Ethical Concerns:**

The paper utilizes standard adaptation settings. Thus no new potential ethical concerns are expected.

**Limitations And Societal Impact:**

Yes, the limitations and impact are discussed.

**Main Review:**

Main review
The main insight adopted in this paper is that as the source samples are not provided, the model embeddings for the target dataset for the initial model (before adaptation) could serve to provide embeddings that are closer to the source domain. Then using these embeddings in a contrastive learning setting, instance-wise and class-wise contrastive learning with these set of models is used to guide the model adaptation procedure.

Pros:
1) The proposed approach is interesting
 2) It is thoroughly  validated with state of the art results for this relatively new task of model adaptation for classification, detection and segmentation.
3) Thorough analysis is provided for the method along with theoretical insights by relating it to EM algorithm


Cons:

While the paper is well written, thoroughly evaluated and with good insights, it has a few issues.

Several aspects of the paper are not very clear and require clarification
1) How are the negative and positive sets identified? One possibility is that for an instance, all historical embeddings for that particular instance form the positive and all other samples would be negative.
2)  If this is the case, negative samples would far outweigh positive samples. How is this imbalance handled.
3) Around line 148 it is mentioned: "For HisNCE, we assign each key a reliability score to encourage to memorize the well-learnt historical embeddings only.” How is this reliability score computed. Later on, for the categorical case, it is mentioned that the samples that are consistent in prediction across current and historical models are termed as well-learnt sample. Is this same criterion used for assigning a reliability score?
4) Similar issue with positive and negative sets for the HCCD learning is also present.

On the whole, the main issues with the paper relate to clarity in writing. There are minor grammatical mistakes also that could be addressed. To conclude, based on the method, analysis and results I retain a positive view on the paper.

Post rebuttal

The rebuttal addresses the concerns raised and provides the clarifications required. I continue to retain my original rating of accept.

**Time Spent Reviewing:**

4

---

> ### Author Response · Authors · 2021-08-10
> **Thank you for your appreciation of our proposed approach, impressive experimental results, and comprehensive analysis with theoretical insights.**
>
> Thank you for your appreciation of our proposed approach, impressive experimental results, and comprehensive analysis with theoretical insights. Below please find our responses regarding your concerns.
>
> Q1: How are the negative and positive sets/pairs defined?
> * We described the definition and formation of positive and negative pairs in Lines 129-134, and they perfectly agree with your ideas. We will further clarify how positive and negative pairs are determined.
>
> Q2: The negative keys far outweigh positive key, how is this imbalance handled?
> * We did not specially handle this imbalance as the larger number of negative samples (negative keys far outweigh positive key) will not degrade the contrastive learning performance. Specifically, most existing contrastive learning methods employ much more negative samples than positive samples. For example, [21] uses $65536$ negative samples versus $1$ positive sample for each query and [9] (as well as our method) uses $4096$ negative samples versus $1$ positive sample. This won’t degrade the contrastive learning performance as the infoNCE loss in contrastive learning is the widely adopted multi-class cross-entropy loss. In fact, prior studies [21,9] showed that a larger dictionary (with more negative keys/samples) leads to better contrastive learning outcome as a larger dictionary with more keys can better represent the dataset distribution. Please refer to the Figure 3 in [21] for more details.
>
>
> Q3: How is this reliability score $r$ computed? Does this paper use the same computation/criterion for the reliability score $r$ and the historical consistency score $h_{con}$?
> * As described in Lines 142-144, we used prediction entropy to compute the reliability score in the proposed historical contrastive instance discrimination (HCID). We will revise relevant text to make it clearer. We use entropy because low entropy usually indicates high confidence (and so reliability) in predictions and vice versa [68,86].
> * The prediction consistency by the historical and current models is instead used to weight the pseudo labels (in self-training) in the proposed historical contrastive category discrimination (HCCD).
>
> Q4: The positive vs. negatives imbalance issue in HCCD?
> * As shared in the response to Q2, we used a standard multi-class cross-entropy loss (in Eq.4) where each sample is pulled closer to the center of a certain class (positive) and pushed apart from the centers of all other $(C-1)$ classes (negatives). The multi-class cross-entropy loss has been widely used in various classification tasks and it does not introduce imbalance issues regardless the number of classes.
> * The imbalance issue is often introduced by the training dataset that consists of an imbalanced class distribution, $e.g.$, one class has much more training samples than another. The proposed historical contrastive category discrimination (HCCD) can mitigate such imbalance with the historical consistency instead of the widely adopted entropy/confidence, as entropy/confidence tends to be biased towards dominant classes in self-training [87,86] (i.e. classes with more training samples often have higher confidence in pseudo labelling). Please refer to Remark 2 (Line 171-178) for more details.

---

### Decision · Program_Chairs · 2021-09-27

**Decision:**

Accept (Poster)

**Comment:**

The paper tackles a challenging problem of unsupervised domain adaptation (UDA) without access to source data while assuming access to a pretrained source model. The proposed method takes inspiration from contrastive losses (such as InfoNCE) and tailors these for the UDA setting. Reviewers appreciated the significance of the problem tackled by the paper (source-free UDA) and the experiments on a variety of vision tasks that show clear improvement over existing methods. Authors should revise the paper taking into account the reviewers' comments, including a discussion of the prior work pointed out in the reviews on the source-free UDA and universal DA problems.